# Folliculotropic Mycosis Fungoides Is Associated with Decreased PD1 Staining Compared with Classic Mycosis Fungoides

Haiming Tang [1,*] , Kristin J. Rybski [2], Yi Luan [3] and Bruce R. Smoller [2]

1 Department of Pathology, Yale School of Medicine, New Haven, CT 06510, USA
2 Department of Pathology and Laboratory Medicine, University of Rochester, Rochester, NY 14642, USA
3 Department of Pharmacology, Yale School of Medicine, New Haven, CT 06510, USA
* Correspondence: haiming.tang@yale.edu

**Abstract:** Programmed cell death protein 1 (PD-1) plays a pivotal role in immune system regulation, with its expression levels linked to malignancy prognosis. However, existing reports on PD-1 staining in mycosis fungoides (MF) present conflicting findings, and little attention has been given to PD-1 staining in different MF variants. To address this, we conducted a retrospective study, employing immunohistochemistry to examine PD-1 expression in cases of folliculotropic MF and non-folliculotropic MF. We analyzed 24 cases of folliculotropic MF and 18 cases of non-folliculotropic MF, and recorded both the percentage of PD-1-labeled tumor cells and the intensity score (negative, weak, medium, or strong). Our results revealed significant disparity in PD-1 labeling between patch/plaque MF and folliculotropic MF ($p = 0.028$). Non-folliculotropic MF exhibited higher PD-1 labeling in tumor cells (58.3%) compared to folliculotropic MF (40.2%). Notably, there was no significant difference in PD-1 staining between folliculotropic MF and non-folliculotropic MF when both were in the early stage/indolent disease category. However, when considering the tumor stage, folliculotropic MF exhibited PD-1 staining in tumor cells at a rate of 21.1%, while non-folliculotropic MF showed PD-1 staining in tumor cells at a rate of 46.6% ($p = 0.005$). Additionally, among folliculotropic MF cases, 13 out of 24 cases displayed differing PD-1 expression patterns between epidermal and dermal components, with preserved PD-1 staining in the epidermal component and loss of staining in the dermal component. Furthermore, consistent with the prior literature, tumor cells with large cell transformations exhibited significantly lower PD-1 labeling ($p = 0.017$). Our findings showcase the unique PD-1 staining patterns in MF.

**Keywords:** PD-1; mycosis fungoides; MF; folliculotropic MF; large cell transformation

## 1. Introduction

Programmed cell death protein 1, recognized as PD-1 and CD279 (cluster of differentiation 279), is a protein present on the surface of T and B cells. It plays a crucial role in modulating the immune response in human cells by dampening the immune system's activity and fostering self-tolerance through the suppression of T-cell inflammation. PD-1 inhibitors, which are a novel category of medications inhibiting PD-1, stimulate the immune system to target tumors and find application in treating specific cancer types.

Previous reports regarding PD-1 staining for mycosis fungoides are somewhat conflicting. Some studies have shown positive PD-1 staining for a large subset (60%) of mycosis fungoides, suggesting that PD-1 could be a differential maker for other cutaneous T-cell lymphomas (usually PD-1 negative) [1]; other studies have shown <15% of mycosis fungoides was positive for PD-1 [2]. One study reported PD-1 positivity in 21 of 25 (84.0%) MF cases and in 11 of 24 (45.8%) other CTCL cases [3]. Another study reported that expression of at least three of five T-follicular helper markers (PD-1, CXCL-13, ICOS, Bcl-6, and CD10) in >10% of tumor cells was observed in 33 out of 36 biopsies [4]. On the contrary, another study suggested that PD-1 expression in MF was so variable that it showed no difference with benign inflammatory dermatoses [5].

Reports of PD-1 expression in different stages of MF are also conflicting. Pileri et al. reported a trend of increasing PD-1 expression in advanced stages of MF. Seven out of eight stage IIIA/B MF patients were shown to have PD-1 expression, while eight of thirteen stage IA/B patients and seven of sixteen stage IIB patients had variable PD-1 positivity [6]. In contrast, Kantekure et al. reported less expression of PD-1 at the tumor stage of MF compared with patch and plaque stage [7].

Large cell transformation has been reported with a loss of PD-1 staining [3,6]. Additionally, it has been reported that Sezary syndrome can have higher PD-1 expression compared with MF, with up to 89% PD-1 positivity [2,8].

Given the conflicting results of PD-1 in MF and the fact that there have not been thorough investigations of PD-1 staining in different variants of MF, it may be of critical value to investigate PD-1 in MF and their variants.

Folliculotropic mycosis fungoides (FMF) is a rare subtype of cutaneous T-cell lymphoma characterized by the infiltration of malignant T-cells into hair follicles, leading to hair loss and often more aggressive clinical behavior compared to classic non-folliculotropic mycosis fungoides (MF). Unlike classic MF, FMF typically presents with follicular papules, pustules, and alopecia, making it challenging to diagnose and manage. The distinctive histopathological feature of FMF is the infiltration of atypical T-cells within the hair follicles, and it tends to have a less favorable prognosis than classic MF, often requiring more aggressive treatment approaches.

In this project, we aim to investigate PD-1 staining in different variants of MF, especially folliculotropic MF. We collected cases with diagnoses of folliculotropic MF as well as matching cases of classical non-folliculotropic MF and compared the PD-1 staining patterns in these cases.

## 2. Methods

Following institutional review board (IRB) approval, this retrospective study was conducted to investigate folliculotropic variants in mycosis fungoides (MF) and classic non-folliculotropic MF cases diagnosed between 1 January 2002 and 1 January 2022. Cases were retrieved from our system database, and all diagnoses were verified and confirmed by experienced pathologists. The study cohort consisted of 24 cases of folliculotropic MF and 18 cases of non-folliculotropic MF.

The clinical data, including age, gender, biopsy site, and clinical stage of MF, were collected for each patient. Specifically, the clinical stage was recorded as early stage/indolent disease vs. tumor stage/aggressive disease. Moreover, we re-evaluated all cases for evidence of large cell transformation. Additional immunohistochemistry (IHC) staining performed at the time of diagnosis, such as CD4, CD8, CD3, CD7, and/or CD30, was also reviewed to complement the analysis.

Immunohistochemistry staining with PD-1 was performed on all cases (Cell Marque PD1 (NAT105), diluted 1:100). The assessment included recording the percentage of tumor cells labeled with PD-1, and assigning a PD-1 intensity score on a scale of negative, weak, medium, or strong. The assessment of the PD-1 tumor cell percentage was performed on tumor cells from all parts of the biopsy. The tumor cells were not counted manually; instead, a percentage of the tumor cells labeling with PD-1 was estimated. The PD-1 assessment was performed independently by two pathologists and consensus was reached.

## 3. Results

A total of 24 cases of folliculotropic MF and 18 cases of non-folliculotropic MF were collected for analysis. The mean age of the patients was 61.9 years (median 63.5, range 25-84). Among all the MF cases, 28 cases were males, and 14 cases were females. Among the 24 cases of folliculotropic MF, 20 cases were males, while 10 of the 18 cases of non-folliculotropic MF were females. Significant differences were observed in the percentage of tumor cells labeled with PD-1 between folliculotropic MF and non-folliculotropic MF ($p = 0.028$). The estimated mean percentage of PD-1 labeling in non-folliculotropic MF was

58.3%, whereas folliculotropic MF showed a lower mean of 40.2% of tumor cells labeled with PD-1. The statistics are shown in Figure 1 and figures of representative cases are shown in Figure 2A–F.

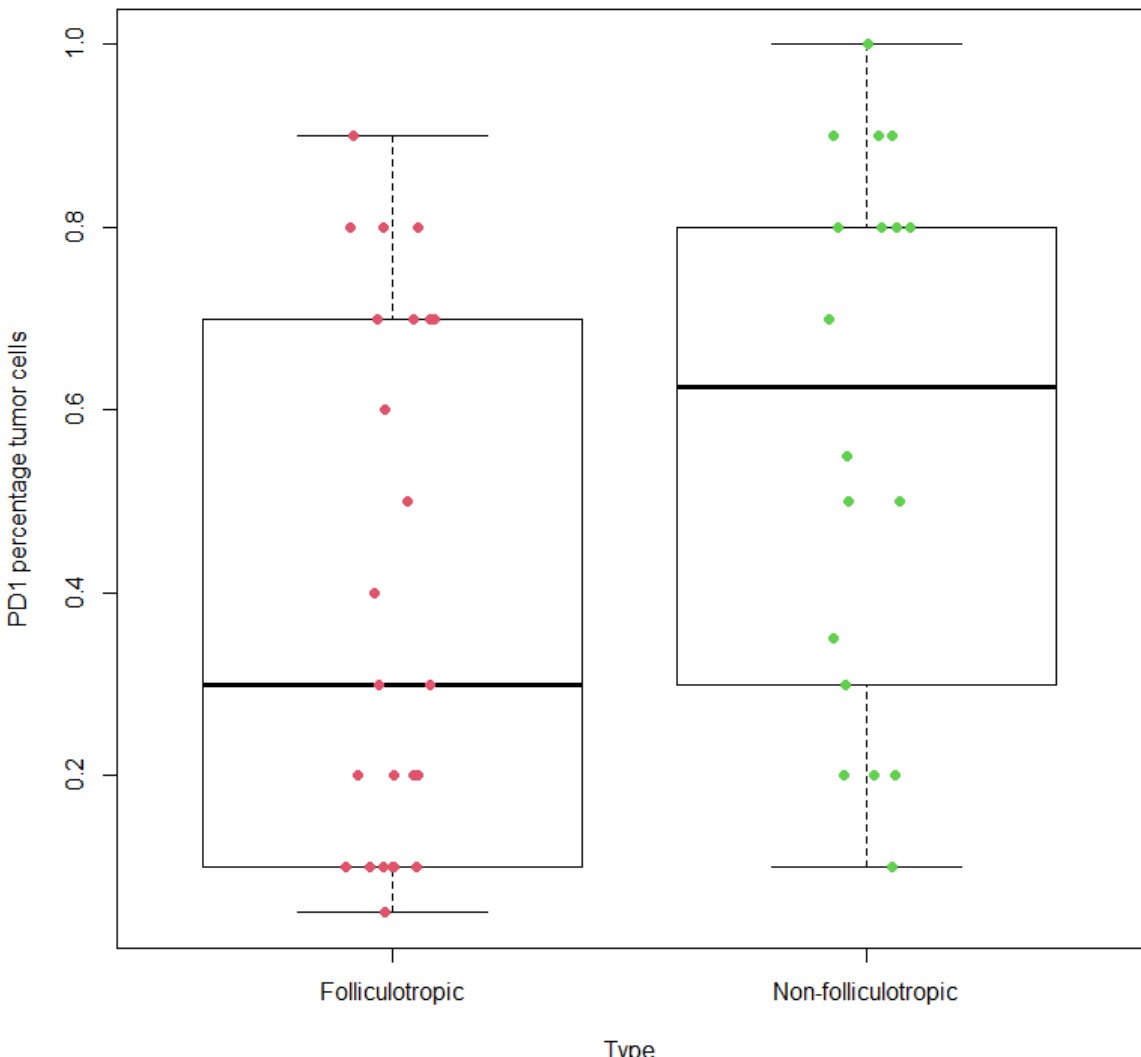

**Figure 1.** Percentage of PD1 staining in tumor cells is statistically significantly lower in folliculotropic MF compared with non-folliculotropic MF, *p*-value = 0.02789.

Among the folliculotropic MF cases, 13 of the 24 cases showed differential expression of PD-1 between the epidermal and dermal components, with a higher percentage of intra-epidermal tumor cells expressing PD-1. The mean difference in percentage of PD-1 labeling between the two components was 0.45 (median 0.50, range from 0.10 to 0.70). A representative case is shown in Figure 2G–I. In cases without differential PD-1 expression, the overall percentage of PD-1 labeling was significantly higher compared to cases with differential expression (*p* = 0.0038, mean of 0.57 vs. 0.26).

Among the 24 cases of folliculotropic MF, 13 cases were indicative of tumor stage/aggressive disease, exhibiting an average PD-1 tumor cell percentage of 21.1%. In contrast, 11 cases represented early stage/indolent disease, displaying an average PD-1 tumor cell percentage of 62.7% (*p* < 0.001). Consequently, it becomes clear that the decrease in PD-1 staining is most pronounced in the advanced/aggressive folliculotropic MF cases.

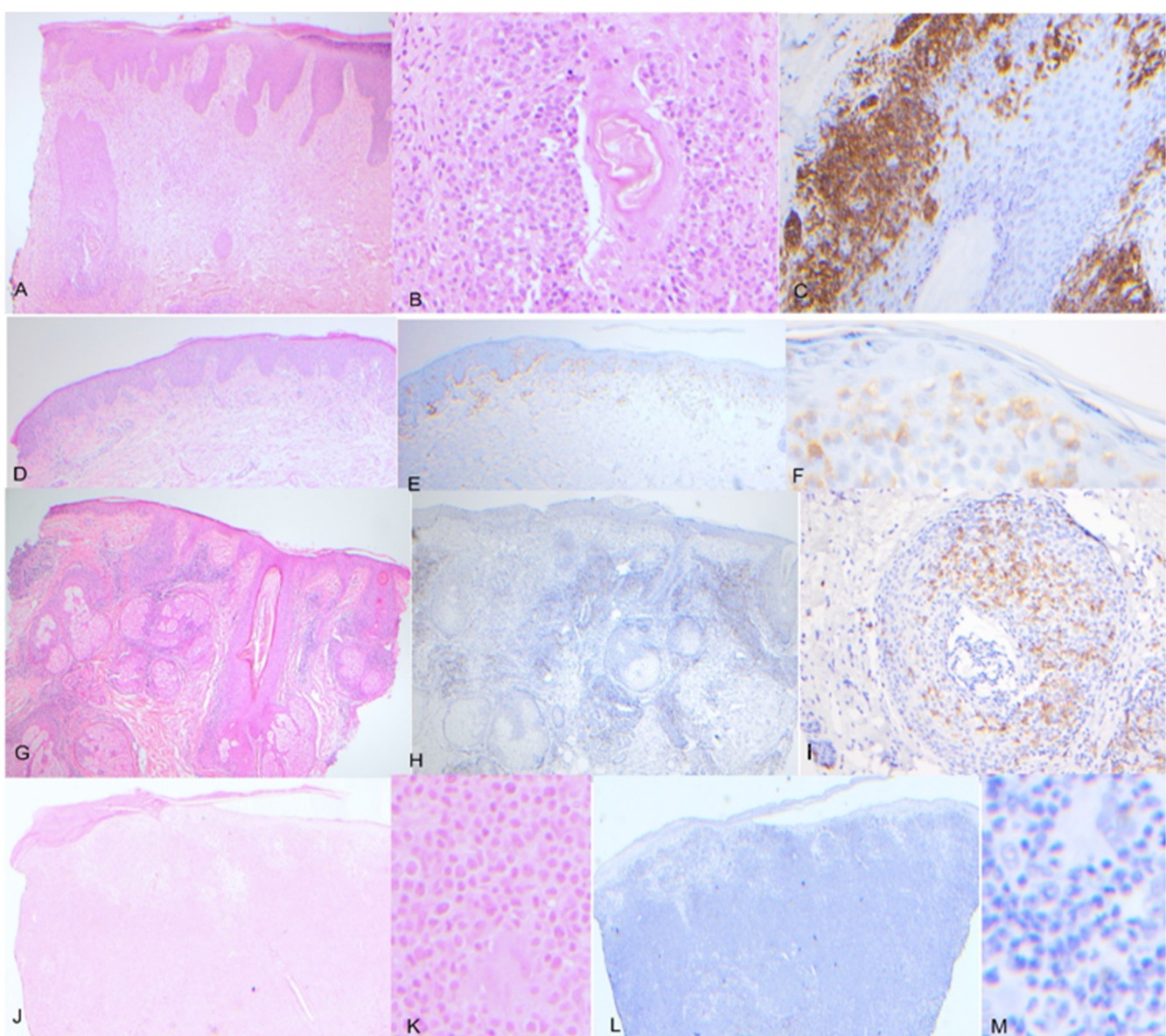

**Figure 2.** Representative cases of folliculotropic MF and non-folliculotropic MF:(**A–C**) Representative folliculotropic MF, showing atypical dense aggregate of lymphocytes preferentially involving the follicle with infiltration to the follicular epithelium, and PD-1 shows strong staining of the tumor cells; (**D–F**) representative non-folliculotropic MF, showing atypical lymphocytes preferentially involving the epidermis, the tumor cells are nicely highlighted by strong PD-1 staining; (**G–I**). representative folliculotropic MF, showing differential PD-1 staining, with retained staining in the epidermal component as in subfigure (**I**) and loss in dermal components in (**H,J–M**). Representative large cell transformation of MF, showing large atypical lymphocytes with extensive dermal involvement. The tumor cells are largely negative for PD-1 staining. ((**A**), H&E 2 × 10; (**B**) H&E 20 × 10; (**C**), PD-1 20 × 10; (**D**), H&E 2 × 10; (**E**) H&E 20 × 10; (**F**), PD1 20 × 10; (**G**), H&E 2 × 10; (**H**), PD-1 2 × 10; (**I**), PD-1 20 × 10; (**J**), H&E 2 × 10; (**K**) H&E 20 × 10; (**L**), PD-1 2 × 10; (**M**), PD-1 20 × 10).

We extended our analysis to further explore variations in PD-1 staining, taking into consideration both the distinction between folliculotropic MF and non-folliculotropic MF, as well as the differentiation between early stage/indolent disease and tumor stage. The results indicate that there is no statistically significant difference in PD-1 staining between folliculotropic MF and non-folliculotropic MF when both are in the early stage/indolent disease category. However, when considering the tumor stage, folliculotropic MF exhibits

PD-1 staining in tumor cells at a rate of 21.1%, while non-folliculotropic MF shows PD-1 staining in tumor cells at a rate of 46.6% ($p = 0.005$).

Regarding large cell transformation, eight cases were identified (five cases in folliculotropic MF and three cases in non-folliculotropic MF). Cases with large cell transformation showed a significantly lower percentage of PD-1 labeling compared to other cases ($p = 0.017$, mean of 0.275 vs. 0.528). A representative case is shown in Figure 2J–M.

CD30 staining was performed in only 20 cases (13 cases in folliculotropic MF and 7 cases in non-folliculotropic MF), and 11 of the 20 cases demonstrated variable degrees of PD-1 expression (five cases in folliculotropic MF and six cases in non-folliculotropic MF). The analysis did not reveal a significant difference in PD-1 labeling based on CD30 status ($p = 0.37$).

CD7 staining was performed in 31 of the 42 cases, with complete loss observed in four cases (all folliculotropic MF) and partial loss in seven cases (four cases in folliculotropic MF and three cases in non-folliculotropic MF). There was no significant difference in PD-1 labeling based on CD7 status ($p = 0.07$).

No statistically significant differences were found in PD-1 staining intensity between folliculotropic MF and non-folliculotropic MF ($p = 0.62$).

## 4. Discussion

In this study, we investigated the PD-1 staining patterns in mycosis fungoides (MF), specifically comparing the folliculotropic variant with the non-folliculotropic MF. The existing literature on PD-1 staining in MF has been inconsistent, with reported positive staining percentages ranging widely from 15% to 84%. To address this variability, we chose not to set a specific threshold for positive or negative PD-1 staining but, instead, estimated the percentage of tumor cells staining positive and the intensity of staining.

It is important to acknowledge that interpretations of immunohistochemistry staining can vary among different pathologists. To minimize bias, we employed a combined approach, considering morphology from H&E slides and the distribution of tumor cells from additional immunohistochemistry staining such as CD3, CD4, CD8, CD7 and CD30. Moreover, the estimates were performed independently by two pathologists and a consensus was reached to enhance accuracy.

A significant finding in this study is the lower percentage of PD-1 staining observed in folliculotropic MF compared to classic non-folliculotropic MF. Additionally, our results suggest a potential interplay between folliculotropic MF and the clinical stage of MF, showing that the decrease in PD-1 staining is most pronounced in the advanced/aggressive folliculotropic MF cases. One study demonstrated that PD-1 expression by tumor cells is more frequent in advanced stage MF and Sezary syndrome (SS) than limited patch/plaque disease [9], which further emphasizes the need for further investigation of the interplay between folliculotropic MF and the clinical stage of MF with a larger dataset. As far as we are aware, this is a novel discovery that has not been previously reported in the literature. Additionally, the presence of differential expression patterns of PD-1, with higher and retained staining of lymphocytes in the epidermal component compared to the dermal component, is a unique observation without clear implications.

The significant differences in PD-1 expression patterns observed between the folliculotropic and classic non-folliculotropic MF may hold important clinical implications. The lower percentage of PD-1 labeling in folliculotropic MF, coupled with its known association with a worse prognosis, raises intriguing questions about the potential role of diminished PD-1 expression in contributing to its more aggressive clinical course. Similarly, the reduced PD-1 labeling observed in cases with large cell transformation, which is also linked to poorer outcomes in MF, suggests a potential interplay between PD-1 expression and disease progression. These findings hint at the possibility that alterations in the PD-1 immune checkpoint pathway might play a role in the development and prognosis of these specific MF variants.

While the exact mechanisms underlying the associations between PD-1 expression patterns and worse prognoses in folliculotropic MF and large cell transformation remain to be elucidated, it is possible that diminished PD-1 expression may lead to a compromised immune response against tumor cells. PD-1 expression levels have been associated with outcomes in other malignancies [10], such as diffuse large B-cell lymphoma, follicular lymphoma, and chronic lymphocytic leukemia, and we speculated on potential associations between reduced PD-1 staining and disease progression or worse outcomes in MF. It is important to note, however, in most malignancies, PD-1 expression was quantified in the tumor microenvironment (TILs) and not the tumor cells themselves, where expression or lack of expression in TILs vs. tumor cells may have very distinct biologic implications. PD-1's role in downregulating T-cell inflammatory activity and promoting self-tolerance suggests that reduced PD-1 expression could result in less effective immune surveillance against malignant cells, allowing for disease progression and worse outcomes. However, further studies are necessary to validate these speculations and unravel the complex interactions among PD-1 expression, the tumor microenvironment, and disease behavior in MF.

Despite the valuable insights gained from this study, certain limitations need to be acknowledged. The relatively small sample size and retrospective nature of the study warrant caution in generalizing the findings. Therefore, further prospective studies with larger cohorts are essential to validate our results and draw more definitive conclusions.

## 5. Conclusions

In conclusion, our study provides valuable insights into the PD-1 expression patterns in different variants of MF, specifically highlighting the significantly lower percentage of PD-1 labeling in folliculotropic MF compared to non-folliculotropic MF. The observed associations with large cell transformation and the differential PD-1 expression pattern in the epidermal and dermal components further add to the complexity of PD-1's role in MF. These findings underscore the potential significance of PD-1 as a prognostic factor in MF overall survival and suggest its relevance in guiding personalized treatment strategies. Further investigations with larger cohorts are warranted to validate these results and to explore the therapeutic implications of PD-1 expression in MF. Ultimately, understanding the intricate relationships between PD-1 and MF subtypes may pave the way for more targeted and effective therapeutic interventions for MF patients.

**Author Contributions:** Conceptualization, H.T. and B.R.S.; methodology, H.T., B.R.S. and K.J.R.; software, H.T; validation, H.T, B.R.S. and Y.L; formal analysis, H.T.; investigation, H.T. and K.J.R.; resources, B.R.S.; data curation, H.T and K.J.R.; writing—original draft preparation, H.T.; writing—review and editing, H.T., B.R.S., Y.L. and K.J.R.; visualization, H.T.; supervision, B.R.S.; project administration, B.R.S.; funding acquisition, NA. All authors have read and agreed to the published version of the manuscript.

**Funding:** This research received no external funding.

**Institutional Review Board Statement:** The study was conducted in accordance with the Declaration of Helsinki, and approved by the Institutional Review Board (or Ethics Committee) of University of Rochester (protocol code STUDY00008131 and date of approval 4 April 2023) and Yale University (protocol code 20000034916 and date of approval 29 March 2023).

**Informed Consent Statement:** Patient consent was waived due to no direct contact with the patients and the study only includes completed/finalized cases from specimens which have been collected and analyzed for diagnostic purposes. Subjects will not be provided with additional information, as they will not be contacted. There is no further information or tissue specimens that will be required of any of the subjects used for this study. The PI also has no way to do this, as he has never had any direct contact with any of the study subjects. The study cannot be completed without a waiver. Many of the patients will have had no clinical follow-up and may not even have been seen as clinical patients at URMC. We will only be looking at tissue taken from biopsies that have been stored in archives. The cases may be many years old. The PI of the study would be unknown to all of the patients.

**Data Availability Statement:** No new data were created or analyzed in this study. Data sharing is not applicable to this article.

**Acknowledgments:** We would like to express our sincere gratitude to the American Society of Dermatopathology (ASDP) for sponsoring this research project through the mentorship award for the year 2023–2024. Their support has been instrumental in the successful completion of this study. Additionally, we would like to thank the ASDP for providing us with the opportunity to present our findings at the ASDP annual meeting in 2023. We extend our appreciation to all the individuals and institutions involved in making this project a reality.

**Conflicts of Interest:** The authors declare no conflict of interest.

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
