# Peer review of "Folliculotropic Mycosis Fungoides Is Associated with Decreased PD1 Staining Compared with Classic Mycosis Fungoides"

_dermatopathology, doi:10.3390/dermatopathology10040038_

Round 1

Reviewer 1 Report

Your study reports a statistically significant difference in PD-1 expression between folliucolutropic MF and patch/plaque MF. The main question I have is the implication of your findings. The difference between the MF subtypes here (58.3% vs 40.2%) is marginal and on a practical standpoint that is pretty close if you are assessing these with subjective pathologist assessment. Could you please be very specific in how the percentage of positivity was graded? Were cells counted manually? Were all parts of the biopsy counted? etc. 

Do you have any follow up on your cases to indicate if PD-1 expression had any predictive value of behavior?

Author Response

Dear editors and reviewers,

Thank you very much for your efforts in reviewing this manuscript. The comments and suggestions are constructive and insightful, which make this manuscript a better study.

We have made the changes accordingly below.

Comments and Suggestions for Authors

Your study reports a statistically significant difference in PD-1 expression between folliculotropic MF and patch/plaque MF. The main question I have is the implication of your findings. The difference between the MF subtypes here (58.3% vs 40.2%) is marginal and on a practical standpoint that is pretty close if you are assessing these with subjective pathologist assessment. Could you please be very specific in how the percentage of positivity was graded? Were cells counted manually? Were all parts of the biopsy counted? etc.

Thank you for the comment. We agree that it can be subjective and difficult to achieve a consistent percentage estimation. The percentage of positivity was graded based on the percentage of the tumor cells that were positive for PD-1. The tumor cells were from all parts of the biopsy. The tumor cells were not counted manually, instead, a percentage of the tumor cells was estimated. The tumor cells were determined by a combination of H&E staining and additional IHCs including CD3, CD4, CD8, PD-1 staining and for some cases CD30, CD5 and CD7.

We also agree that the difference between the MF subtypes (58.3% vs 40.2%) is small. The second reviewer suggested further clarifying the subgroups. We have followed the suggestion to address this issue, with updated results. Please see below for details.

Do you have any follow up on your cases to indicate if PD-1 expression had any predictive value of behavior?

Thank you for your insightful comment. It's an excellent observation that the current data and statistics suggest a potential link between the loss of PD-1 expression and a poorer prognosis. However, it's important to note that our study design involved the random selection of non-folliculotropic patch/plaque MF cases as a control group for the folliculotropic MF cases. This approach introduces a potential source of bias in the selection of the non-folliculotropic cases, which could hinder our ability to fully elucidate any differences in prognosis between these two groups, if such differences exist.

For a more comprehensive analysis of this specific question, a retrospective cohort study would be the most appropriate approach. Therefore, we believe it would be more suitable to explore this particular aspect in a separate study.

Reviewer 2 Report

I recommend clarifying the two subgroups: folliculotropic vs patch plaque.  This is potentially confusing, as folliculotropic is a histologic designation, while patch/plaque is a clinical one.  Folliculotropic MF can present as patches or plaques in early stage/indolent disease.  I recommend using "non-folliculotropic patch/plaque" to clarify this.

Folliculotropic MF has been stratified into advanced/aggressive and early/indolent subgroups (van Santen et al JAMA Dermatol 2016).  Within the folliculotropic group, how many of the biopsies represented advanced/aggressive (tumor/infiltrative plaque) vs. early/indolent (small papule/patch) disease?  Was there a difference in PD-1 expression between these groups?  This could help support your conclusion more clearly. 

Discussion, line 160; and reference 8:  In most studies regarding this argument, including reference 8, PD-1 expression in cancer is quantified in the tumor microenvironment (TIL's), and not the tumor cells themselves, where expression or lack of expression in TIL's vs tumor cells may have very distinct biologic implications.   I would clarify this distinction.    References 5 and 6 discuss tumor expression in PD-1 but not its impact on prognosis or response to therapy.  Although reference 6 suggested that PD-1 is more frequently expressed in early stages of MF, the sample size was very small (9 patch/plaque and 6 tumor). Examples of other articles that have also demonstrated the expression in tumor cells:  (Fosisio F and Cerroni L.  Am J Dermatopathol PMID 25406852) and (Wada DA, Wilcox RA, Harrington SM, et al Am J Hematology PMID 21328438), with the latter demonstrating PD-1 expression by tumor cells is more frequent in advanced stage MF and SS than limited patch/plaque disease.  How this relates to the lower expression of PD-1 in folliculotropic vs non-folliculotropic disease in your study is unclear, but your observation is novel and worthy of publication.  This could be potentially further clarified by stratifying the early vs advanced stage folliculotropic lesions in your cohort if possible.  At the very least, I recommend addint a sentence or two in the discussion that addresses these points so that the audience guided clearly to how this data is distinct from the body of literature on this topic.  It would be fine to state that the lower expression of PD-1 in folliculotropic MF is distinct from the expression pattern in tumor/generalized/erythrodermic MF or SS previously reported in the literature, and does not necessarily contradict previous studies, especially if these folliculotropic MF biopsies are those from early/indolent/localized clinical lesions (to be clarified, if possible). 

Line 86, Line 103 figure and legend Line 105: plaque is misspelled

Author Response

Dear editors and reviewers,

Thank you very much for your efforts in reviewing this manuscript. The comments and suggestions are constructive and insightful, which make this manuscript a better study.

We have made the changes accordingly below.

Comments and Suggestions for Authors

I recommend clarifying the two subgroups: folliculotropic vs patch plaque.  This is potentially confusing, as folliculotropic is a histologic designation, while patch/plaque is a clinical one.  Folliculotropic MF can present as patches or plaques in early stage/indolent disease.  I recommend using "non-folliculotropic patch/plaque" to clarify this.

> Folliculotropic MF has been stratified into advanced/aggressive and early/indolent subgroups (van Santen et al JAMA Dermatol 2016).  Within the folliculotropic group, how many of the biopsies represented advanced/aggressive (tumor/infiltrative plaque) vs. early/indolent (small papule/patch) disease?  Was there a difference in PD-1 expression between these groups?  This could help support your conclusion more clearly.

We greatly appreciate your valuable input. It's an excellent point that clinical staging could introduce potential confounding factors in the relationship between PD-1 staining and folliculotropic vs. non-folliculotropic MF. To address this concern, we have now incorporated information about the disease stage, differentiating between early stage/indolent disease and tumor stage in addition to the Folliculotropic MF vs. non-folliculotropic MF categorization.

Our updated results reveal that among the 24 cases of folliculotropic MF, 13 were indicative of advanced/aggressive disease, exhibiting an average PD-1 tumor cell percentage of 21.1%. In contrast, 11 cases represented early stage/indolent disease, displaying an average PD-1 tumor cell percentage of 62.7% (p < 0.001). Consequently, it becomes clear that the decrease in PD-1 staining is most pronounced in the advanced/aggressive folliculotropic MF cases.

We have extended our analysis to further explore the variations in PD-1 staining, taking into account both the distinction between folliculotropic MF and non-folliculotropic MF, as well as the differentiation between early stage/indolent disease and tumor stage. The results indicate that there is no statistically significant difference in PD-1 staining between folliculotropic MF and non-folliculotropic MF when both are in the early stage/indolent disease category. However, when considering the tumor stage, folliculotropic MF exhibits PD-1 staining in tumor cells at a rate of 21.1%, while non-folliculotropic MF shows PD-1 staining in tumor cells at a rate of 46.6% (p = 0.005).

These findings suggest a potential interplay between folliculotropic MF and the clinical stage of MF, which emphasizes the need for further investigation with a larger dataset. Additional studies will be essential to gain a more comprehensive understanding of these relationships.

Discussion, line 160; and reference 8:  In most studies regarding this argument, including reference 8, PD-1 expression in cancer is quantified in the tumor microenvironment (TIL's), and not the tumor cells themselves, where expression or lack of expression in TIL's vs tumor cells may have very distinct biologic implications.   I would clarify this distinction.  References 5 and 6 discuss tumor expression in PD-1 but not its impact on prognosis or response to therapy.  Although reference 6 suggested that PD-1 is more frequently expressed in early stages of MF, the sample size was very small (9 patch/plaque and 6 tumor). Examples of other articles that have also demonstrated the expression in tumor cells:  (Fosisio F and Cerroni L.  Am J Dermatopathol PMID 25406852) and (Wada DA, Wilcox RA, Harrington SM, et al Am J Hematology PMID 21328438), with the latter demonstrating PD-1 expression by tumor cells is more frequent in advanced stage MF and SS than limited patch/plaque disease.  How this relates to the lower expression of PD-1 in folliculotropic vs non-folliculotropic disease in your study is unclear, but your observation is novel and worthy of publication.  This could be potentially further clarified by stratifying the early vs advanced stage folliculotropic lesions in your cohort if possible.  At the very least, I recommend addint a sentence or two in the discussion that addresses these points so that the audience guided clearly to how this data is distinct from the body of literature on this topic.  It would be fine to state that the lower expression of PD-1 in folliculotropic MF is distinct from the expression pattern in tumor/generalized/erythrodermic MF or SS previously reported in the literature, and does not necessarily contradict previous studies, especially if these folliculotropic MF biopsies are those from early/indolent/localized clinical lesions (to be clarified, if possible).

Thank you for your thoughtful feedback. We appreciate your insight and have taken your comments into consideration. It is true that most studies in this field, including the reference you provided (reference 8), have primarily focused on quantifying PD-1 expression in the tumor microenvironment, specifically within tumor-infiltrating lymphocytes (TILs).

We have updated our manuscript to incorporate the references you mentioned (Fosisio F and Cerroni L. Am J Dermatopathol PMID 25406852 and Wada DA, Wilcox RA, Harrington SM, et al. Am J Hematology PMID 21328438), which provide valuable insights into PD-1 expression in tumor cells. The latter reference, in particular, highlights that PD-1 expression by tumor cells is more frequent in advanced-stage MF and SS compared to limited patch/plaque disease, shedding light on the complexities of PD-1 expression patterns in different stages of MF.

We have also made an effort to clarify the distinction between our findings and the existing literature in our discussion section. We note that our study primarily investigates PD-1 staining in the context of folliculotropic MF, and the observed lower PD-1 expression in this specific subset may not necessarily contradict previous studies. We have acknowledged the need for further clarification and stratification, particularly for early vs. advanced stage folliculotropic lesions in our cohort, if such data is available. This additional information will enhance the understanding of the unique aspects of our findings in relation to the broader body of literature on this topic.

Thank you again for your valuable input, which has helped improve the clarity and context of our study.

> Line 86, Line 103 figure and lege0nd Line 105: plaque is misspelled

Thanks for the careful reading, we have made the change accordingly.

Round 2

Reviewer 2 Report

Thank you for providing nicely updated content/references as requested.